# Novel Application of the Lagis LapBase Cap in Transvaginal NOTES Hysterectomy: Surgical Outcomes and Cost-Effectiveness in 107 Cases

**DOI:** 10.3390/jcm14217796

**Published:** 2025-11-03

**Authors:** Yu-Tung Hsieh, Shi-Bei Liang, Yu-Fang Hsu, Chun-Shuo Hsu

**Affiliations:** 1Department of Obstetrics and Gynecology, Dalin Tzu Chi Hospital, Buddhist Tzu Chi Medical Foundation, Chiayi 622, Taiwan; 2Taichung Veterans General Hospital, Taichung 407, Taiwan; 3Institute of Medicine, Chung Shan Medical University, Taichung 402, Taiwan; 4School of Medicine, Tzu Chi University, Hualien 970, Taiwan

**Keywords:** Lagis LapBase Cap, hysterectomy, laparoscopy, NOTES, transvaginal surgery

## Abstract

**Objectives:** Transvaginal Natural Orifice Transluminal Endoscopic Surgery (vNOTES) offers significant benefits in hysterectomy, including reduced postoperative pain, minimal scarring, and faster recovery. However, the cost and accessibility of surgical ports can be limiting factors. This study aimed to evaluate the feasibility, outcomes, and cost-effectiveness of using the Lagis LapBase Cap as an alternative port system in 107 vNOTES hysterectomy cases at a single institution. **Methods:** A retrospective analysis was conducted on 107 patients who underwent vNOTES hysterectomy between January 2017 and April 2022. Patients with benign gynecologic conditions and no suspected malignancy or deep infiltrating endometriosis were included. The Lagis LapBase Cap was used for access via an Alexis wound retractor. Surgical parameters—including operation time, estimated blood loss, and length of hospital stay—were analyzed by uterine weight, BMI, and obstetric history. **Results:** Of the 107 cases, 104 were completed using vNOTES, with only 3 conversions to laparoscopy. The average operation time was 88 min, and the mean estimated blood loss was higher in patients with larger uteri or BMI ≥ 24. Nulliparous women and those with a history of multiple cesarean sections also had longer operation times. There were no major complications, and most patients were discharged within three days postoperatively. **Conclusions:** The Lagis LapBase Cap is a practical and cost-efficient tool for vNOTES hysterectomy. It provides reliable sealing and instrument access, while maintaining favorable surgical outcomes. Patient selection based on uterine size, BMI, and delivery history may help optimize procedural efficiency.

## 1. Introduction

Numerous minimally invasive procedures have been developed for gynecologic surgery, including single-port laparoscopic approach surgery and vaginal Natural Orifice Transluminal Endoscopic Surgery (vNOTES). A lot of effort has been made in improving outcomes for patients. NOTES was first described by Kalloo et al. in 2004 in a porcine model as an extended application of single-port laparoscopic surgery [1]. Since then, many physicians have performed vNOTES surgery and subsequent studies have demonstrated the safety and efficacy of this procedure. vNOTES is a safe and feasible procedure even in large uteri [2].

In 2014, a 137-patient large case series was published and reported a low complication rate of one postoperative voiding disorder and four postoperative fevers [3]. Conversions to traditional laparoscopic hysterectomy in seven patients were mostly due to failure of the transvaginal colpotomy; however, the overall success rate was 94.8%. Compared to traditional laparoscopic hysterectomy, a 2015 study by Wang et al. reported that vNOTES was associated with significantly shorter surgical time, less estimated blood loss, and a shorter length of stay [4]. vNOTES has been shown to be non-inferior to conventional laparoscopy in a randomized trial and may facilitate higher rates of day-care discharge [5] and systematic reviews further support its safety and feasibility in benign gynecologic indications [6].

There are several methods for applying vNOTES port and instruments, including the wound-retractor-and-glove system and the Single Port Access Device, available from different companies [7,8,9]. In our department, we used the Lagis wound retractor with the Lagis LapBase Cap for Wound Retractor 60 mm. In our experience, this device has advantages like ease of use and a good seal. During 2017 to 2022, one hundred and seven patients underwent vNOTES hysterectomy in our institution. In this article, we share the experience of 107 operations, including operation methods and outcomes.

## 2. Materials and Methods

### 2.1. Patients

This study was conducted in accordance with the Declaration of Helsinki, and approved by the Research Ethic Committees of Dalin Tzu Chi Hospital (IRB-B11104014). Informed consent was obtained from all subjects involved in this study.

We enrolled women who underwent vNOTES hysterectomy, regardless of whether adnexa surgery was carried out or not, for the indication of benign gynecologic disease from January 2017 to April 2022 at Buddhist Tzu Chi Medical Foundation Dalin Tzu Chi Hospital. We used the Taiwan national health insurance reimbursement claim code (80416B) to search the electronic medical record system and selected those where NOTES was recorded.

### 2.2. Surgical Technique

At our institution, we performed vNOTES hysterectomy in a non-prolapsed uterus for benign gynecologic disease, including myoma, adenomyosis, and cervical dysplasia. We excluded patients with known or suspected malignancy due to the uterus possibly being resected during retrieval. Preoperative dynamic transvaginal ultrasound assessed the “sliding sign” to screen for pouch of Douglas obliteration, a validated marker of posterior cul-de-sac adhesions and deep endometriosis [10,11]. A transvaginal sonography was performed preoperatively to exclude patients with cul-de-sac adhesion. If sliding signs in transvaginal ultrasound (TVS) were negative, an adhesion was impressed and it was recognized as a contraindication for vNOTES. A history of cesarean section or pelvic surgery was not a contraindication in our department. All procedures were performed by one surgeon, C.S. Hsu.

Patients were placed in the lithotomy position once they were under general anesthesia. Prophylactic antibiotics, with Cephradine 2 g, were administered intravenously preoperatively. A Foley catheter was placed, and we fixed the vulvas outward with silk suture to make it easier for the surgeon to perform the operation. A total of 20U of Vasopressin was diluted in 60 mL normal saline and injected circumferentially into the junction of the cervix and vaginal mucosa for vasoconstriction. We then made an incision around the cervix and performed anterior and posterior colpotomy. The cervicovesical junction was identified and the anterior cul-de-sac was entered; then, the posterior was entered using the same steps. The bilateral uterosacral ligaments and cardinal ligaments were clamped, transected, and sutured and then ligated with 1-0 Vicryl. We then performed cervical amputation and sutured the root with 1-0 Vicryl for better vision and a larger space (Figure 1A).

Afterwards, the Lagis wound retractor 60 mm (Lagis Enterprise Co., Ltd., Taichung, Taiwan) was placed through the anterior colpotomy and the posterior colpotomy. The Lagis LapBase Cap for Wound Retractor 60 mm was applied to seal the wound retractor. A 10 mm trocar was placed at the inferior area (6 o’clock) of the cap to maintain pneumoperitoneum, with an intraperitoneal pressure of 14 mmHg, and then a 30-degree endoscope was inserted. Two 5 mm trocars were inserted at the superior area of the cap, and a Ligasure vessel sealing device was applied into the right trocar and forceps into the left trocar (Figure 1B). Bilateral board ligaments, ovarian ligaments, and fallopian tubes were coagulated and cut (Figure 1C,D). Depending on whether adnexa surgery was planned or not, we then performed salpingectomy or salpingo-oophorectomy. After the uterus was set free, it was removed from the vagina. With a large uterus or one with myomas, the surgeon would resect the uterus with a cold knife. After good hemostasis was confirmed, the vaginal cuff was closed with 1-0 Vicryl.

After the operation, prophylactic antibiotics, with Cephradine 500 mg every 6 h for one day, and a single dose of Gentamycin 240 mg were given. The Foley was removed the day after the operation, or if the surgery was performed in the morning it was removed at night on the same day. The patient was typically discharged on postoperative day 1 or day 2.

### 2.3. Outcome Measures

The primary outcome measures for this study were the successful completion of vNOTES hysterectomy (defined as completion without conversion to another surgical approach), and key intraoperative outcomes, including total operation time (in minutes) and estimated blood loss (EBL) in milliliters (mL).

The secondary outcome measures included the rate of blood transfusion, postoperative change in hemoglobin (Hb), length of hospital stay (in days), and the incidence of intra- and postoperative complications (such as bowel or bladder injuries)

### 2.4. Statistical Analysis

This study was conducted as a retrospective, descriptive case series of the 107 consecutive patients who underwent a vNOTES hysterectomy for benign gynecologic disease. No a priori sample size calculation was performed.

All data were collected from the electronic medical records system. Descriptive statistics were used to summarize patient characteristics and surgical outcomes. Continuous variables (e.g., age, operation time, uterine weight) were presented as mean (range) values. Categorical variables (e.g., parity, BMI group, delivery history) were presented as counts (*n*) and percentages (%). The analysis was stratified by uterine weight (<300 g vs. ≥300 g), BMI (<24 vs. ≥24), and delivery history to describe outcomes within these subgroups.

All analyses were performed using SPSS Statistics Version 25 (IBM Corp., Armonk, NY, USA).

## 3. Results

Between January 2017 and April 2022, there were 104 patients who underwent NOTES hysterectomy in our hospital, and an additional 3 patients underwent NOTES at first but then shifted to conventional laparoscopic-assistance vaginal hysterectomy (LAVH). The patients were aged between 28 and 67 years old and were diagnosed with leiomyoma, adenomyosis, or cervical dysplasia. Based on the final pathologic report, three patients were diagnosed with Squamous cell carcinoma in situ of the cervix, one patient with carcinoma in situ of the cervix and one patient with well-differentiated endometrioid carcinoma. The mean weight of the uterus was 302.3 gm (range 61.2 gm to 1189 gm).

Patients were grouped according to uterine weight and BMI for analysis. Sixty-six patients had a uterine weight of <300 g, and thirty-eight had a uterine weight of ≥300 g. Patient characteristics are listed in Table 1. The mean age of the small uterine group was 44.1 years, and for the big uterine group this was 44.8 years. There were no differences in mean parity and mean BMI between the two groups. Two patients in the small uterine group and three patients in the big uterine group were nullipara; sixteen and ten patients had no history of vaginal birth in the small uterine and big uterine group, respectively. BMI ≥ 24 was defined as overweight, according to the Health Promotion Administration in Taiwan. More patients had a lower BMI in the small uterine group (59.0%), and more had a higher BMI in the big uterine group (55.3%). More patients in the big uterine group had anemia before the operation (28.8% in the small uterine group and 57.9% in the big uterine group). Eight patients in the small uterine group and six patients in the large uterine group had a history of abdominal or pelvic surgery other than cesarean section, including appendectomy, myomectomy, and adnexa surgery.

The mean operation time was 88 min for all patients (81 min in the small uterine group and 98 min in the big uterine group). The mean length of stay was 3.06 days for all patients (2.95 days in the small uterine group and 3.28 days in the large uterine group). In the big uterine group, there was more estimated blood loss (mean: 71 mL vs. 143 mL) and more patients received blood transfusions (7 patients vs. 10 patients) (Table 2).

Stratified by BMI, in the small uterine group, there were 39 patients with BMI < 24 and 27 patients with BMI ≥ 24. There was no difference in operation time, blood loss, and length of stay between patients with a lower and higher BMI. In the big uterine group, there were 17 patients with BMI < 24 and 21 patients with BMI ≥ 24. The patients that had a bigger uterus and a higher BMI had a longer operation time, more estimated blood loss, and a longer length of stay (Table 3).

We hypothesized that vNOTES surgery would be easier on women with a history of vaginal delivery (VD), due to easier colpotomy. Thus, we compared outcomes between women with VD, women without VD but with cesarean section (CS), and nullipara women. The mean operation time was longer in the nullipara women (85, 91, and 110 min, respectively) and mean blood loss was also higher in the nullipara women (94.7, 100 and 130 mL, respectively) (Table 4).

Previous CS might cause adhesion between the bladder and uterus, and lead to longer operation times or more difficulty in performing anterior colpotomy. We found that patients with a history of more than two CS operations had a longer operation time. The mean operation time was 100 min in patients with a history of >2 CS, 90 min in patients with a history of one CS, and 83 min in patients without a history of CS but with a history of VD (Table 4).

Three patients converted to LAVH. One was due to colon adhesion to the uterine fundus, one with adhesion at the cul-de-sac, and one with a narrow vagina.

## 4. Discussion

This is the first study describing the usage of Lagis LapBase Cap in vNOTES surgery. Most vNOTES surgeries were conducted with a single-port device, and some were conducted with a handmade glove port. Based on our experience, the cap worked well in sealing and supporting the surgery instruments. At our hospital, the Lagis LapBase Cap cost around NTD 5500 (about USD 170). It is more cost-effective for the institution and affordable for the patient. The single-port device, on the other hand, typically costs more than NTD 10,000 (USD 310). This is a retrospective study of one hundred and seven patients who underwent vNOTES hysterectomy in Taiwan and who had a low conversion rate of 2.8% (3/107). One hundred and four patients successfully received the vNOTES procedure. There were no bowel or bladder injuries in the 107 patients.

Among the 104 patients successfully receiving the vNOTES procedure, the mean operation time was 88 min, which is comparable to previous studies in Taiwan, Korea, Turkey, and France [3,12,13]. We divided patients into groups according to uterus size and BMI, and tried to identify risk factors for a longer operation time. Wang C.J. showed that longer operations were associated with patients with a bigger uterus [4]. A 2020 Belgian study and a 2025 study by Ryo Chee Ann Tan found similar results [9,10]. A French study published in 2023 discussed operation outcomes. The mean operative time was shorter in the vNOTES group than in the laparoscopy group [116 min versus 149 min; *p* = 0.003] [8]. Longer operative times are associated with higher body mass index (BMI). The median operative time for vNOTES hysterectomy was 79 (44–128) min, 104 (48–193) min, and 106 (45–106) min in obesity class I, II, and III patients, respectively—class I (BMI 30.1–34.9 Kg/m^2^), class II (BMI 35.0–39.9 Kg/m^2^), and class III (BMI ≥ 40.0 Kg/m^2^) [14].

According to our study, patients with a uterus smaller than 300 gm had an average operation time of 81 min. There was no difference in operation time between lower BMI and higher BMI in patients with a small uterus (<300 gm). Among patients with a uterus larger than 300 gm, the average operation time was 98 min, and it took even more time if BMI ≥ 24 kg/m^2^, compared to those with BMI < 24 (average 106 min compared to 89 min). Nulliparity was not a contraindication for the vNOTES procedure; however, we hypothesized that a narrow vagina might cause more difficulty in performing colpotomy, and thus cause a longer operation time or even conversion. The average operation time for nullipara women (*n* = 5) was 110 min. Among those requiring conversion surgery, one patient was nullipara and colpotomy failed due to a narrow vagina. Among women with a history of cesarean section, and without a history of vaginal delivery, a CS incision adhesion at the anterior lower uterus segment also led to a longer operation time. In our experience, we could place a wound retractor after a posterior colpotomy was completed, even when an anterior colpotomy was incomplete due to a CS incision adhesion. Adhesiolysis could be conducted under laparoscopy (Figure 2); however, the adhesion at low segments might cause the wound retractor to slip out of the vagina and it would take time for the surgeon to replace it. According to our data, women with a history of more than two CS experienced longer operation times, with an average of 100 min.

Most patients at our hospital were discharged on postoperative day 2, with a length of stay of 3 days. Some patients who recovered well, with little Hb change, minimal postoperative wound pain, and the ability to ambulate and tolerate oral intake, were discharged the day following the operation. However, the average length of stay for patients with a large uterus was 3.23 days. According to our data, we found that longer operation times and preoperative anemia were also associated with a longer length of stay. There were 41 patients with Hb < 11 before the operation and 16 of them received a blood transfusion. The average length of stay for this group was 3.19 days and 12 were admitted for more than 4 days. Among those without anemia (*n* = 63), the average length of stay was 2.93 days. Only 1 of the 63 patients in this group received a blood transfusion due to postoperative anemia with dizziness. Our study followed an institutional antibiotic protocol that included cefazolin and gentamicin intraoperatively, plus 24 h of cephradine. This differs from current guidelines recommending single-dose prophylaxis and should be considered when interpreting our results [15].

## 5. Conclusions

The Lagis LapBase Cap provided a stable and efficient access platform for transvaginal NOTES hysterectomy across a cohort of 107 patients, with a high procedural success rate (97.2%) and no major complications. Surgical difficulty was most influenced by uterine weight ≥300 g and a history of multiple cesarean sections. Nulliparity was associated with longer operative time and higher blood loss, likely due to limited vaginal exposure and technical challenges during colpotomy. While elevated BMI alone was not predictive of increased complexity, its combination with large uterine size resulted in prolonged surgery. Preoperative anemia demonstrated a stronger association with extended hospitalization than either BMI or uterine weight. These findings suggest that appropriate patient selection and risk stratification are critical in optimizing outcomes of vNOTES hysterectomy. The LapBase Cap may facilitate broader adoption of the technique by offering cost efficiency and technical reliability, particularly in settings where access to commercial single-port devices is limited, similar to other glove-and-retractor-based platforms described in gynecologic laparoscopy [16].

## Figures and Tables

**Figure 1 jcm-14-07796-f001:**
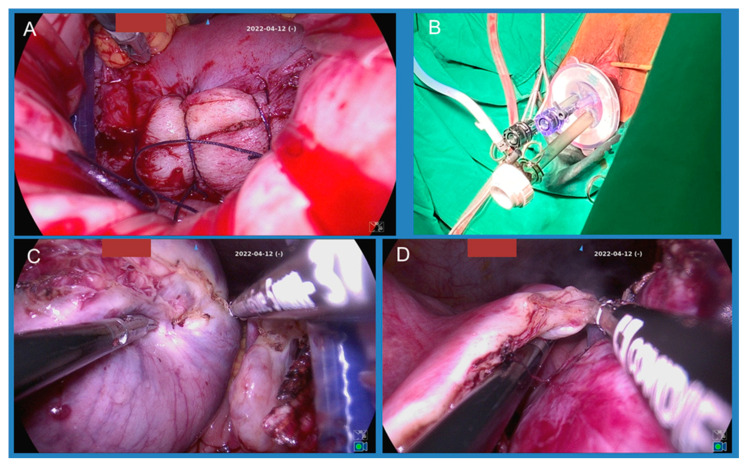
(**A**) Colpotomy and cervical amputation were performed; (**B**) the wound retractor with cap and trocars were placed; (**C**,**D**) bilateral broad ligaments, ovarian ligaments, and fallopian tubes were coagulated and cut.

**Figure 2 jcm-14-07796-f002:**
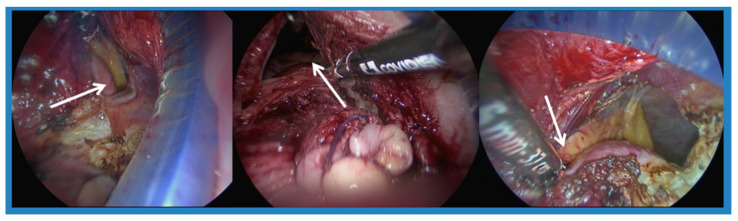
The arrow points to adhesion at the CS wound and the anterior wall of uterus, which made the wound retractor slip out easily.

**Table 1 jcm-14-07796-t001:** Patient characteristics for the large and small uterine groups.

	Uterine Weight < 300 gm (*n* = 66)	Uterine Weight ≥300 gm (*n* = 38)
Age (years)	44.1 (32–55)	44.8 (28–52)
Parity	2 (2.1)	2 (2.0)
nulliparous (*n*)	2	3
Delivery type		
vaginal delivery (*n*)	50 (75.8%)	28 (73.7%)
cesarean section (*n*)	20 (30.3%)	9 (23.7%)
no history of vaginal delivery (*n*)	16 (24.2%)	10 (26.3%)
Body mass index (BMI) (mean)	23.8 (17.3–37.8)	24.8 (18.3–32.1)
BMI < 24 (*n*)	39 (59.1%)	17 (44.7%)
BMI ≥ 24 (*n*)	27 (40.9%)	21 (55.3%)
History of abdominal and pelvic surgery, excluding C/S (*n*)	8	6
Preoperative anemia (Hb < 11) (*n*)	19 (28.8%)	22 (57.9%)

BMI = Body Mass Index.

**Table 2 jcm-14-07796-t002:** Operation outcomes.

	Uterine Weight < 300 gm (*n* = 66)	Uterine Weight ≥ 300 gm (*n* = 38)
Operation time (minutes)	81	98
Estimated blood loss (mL)	71 (minimal–300)	143 (50–400)
Blood transfusion (*n*)	7 (2U)	10 (2–8U)
Hb change (excluding patients needing blood transfusion)	−1.1 (0.3 to −4.6)	−0.9 (−0.1 to −2.4)
Length of stay (days)	2.92	3.23

Hb = Hemoglobin.

**Table 3 jcm-14-07796-t003:** Operation outcomes, sub-stratified by BMI.

	Uterine Weight < 300 gm		Uterine Weight ≥ 300 gm	
	BMI < 24 (*n* = 39)	BMI ≥ 24 (*n* = 27)	BMI < 24 (*n* = 17)	BMI ≥ 24 (*n* = 21)
Operation time (minutes)	82	80	89	106
Estimated blood loss (mL)	79	59	118	164
Blood transfusion (*n*)	3	4	3	7
Hb change (excluding patients needing blood transfusion)	−1.3 (0.1 to −4.6)	−0.8 (0.3 to −2.5)	−0.8 (−0.1 to −2)	−1.0 (−0.1 to −2.4)
Length of stay (days)	3.07	2.70	3	3.42

Hb = Hemoglobin, BMI = Body Mass Index.

**Table 4 jcm-14-07796-t004:** Characteristics of patients based on delivery history.

	CS > 2 times (14)	CS 1 Time (15)	Only VD (70)	Nullipara (5)
Uterine size	336.6	284.0	300.4	287.3
BMI	23.81	24.38	24.04	26.65
Operation times	100	90	83	110
Estimated blood loss (mL)	89.2	92	98	130
Blood transfusion (*n*)	3	1	11	2
Admission days	3	2.86	3.07	3.2

CS = Cesarean Section, VD = Vaginal Delivery, BMI = Body Mass Index.

## Data Availability

The original data can be asked for, upon reasonable request, by contacting the corresponding author.

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
