# Peer review of "Novel Application of the Lagis LapBase Cap in Transvaginal NOTES Hysterectomy: Surgical Outcomes and Cost-Effectiveness in 107 Cases"

_jcm, 2025, doi:10.3390/jcm14217796_

Round 1
Reviewer 1 Report
Comments and Suggestions for Authors
This retrospective observational study evaluates the feasibility, outcomes, and cost-effectiveness of using the Lagis LapBase Cap as an alternative port system in Transvaginal Natural Orifice Transluminal Endoscopic Surgery (vNOTES) hysterectomy cases at a single institution. The paper is well written, and the English language is appropriate and understandable. The clinical arguments presented are interesting due to the increasing relevance of vNOTES that could offer significant benefits in hysterectomy for benign disease, including reduced postoperative pain, minimal scarring, and faster recovery. To date, data on the safety and efficacy of vNOTES are well-documented in the literature, with a low rate of conversions to conventional laparoscopy. However, the costs represent the major drawbacks of this surgical technique. The data from this study support the role of vNOTES in terms of feasibility, yielding better results without major complications in the selected setting of patients with smaller uteri, lower BMI, and no history of cesarean sections. However, although the statistical analysis was well performed, this is a retrospective study with multiple biases that limit its clinical validity. Furthermore, considering the potential indication of vNOTES for the treatment of benign gynecologic lesions, including myoma, adenomyosis, and cervical dysplasia, the simple size is quite small, especially in the analysis of subgroups of patients stratified by BMI and uterus weight. Specific comments: The Author reported that the patient was typically discharged on post-operative day 1 or day 2. However, the results showed that the mean length of stay was 3.06 days for all patients (2.95 days in the small uterine group and 3.28 days in the large uterine 133 group). Was cervical amputation always performed in case of a smaller uterus? The Authors reported that one of the major outcomes of this study was the evaluation of the cost-effectiveness of using an alternative port system. Could the Authors provide more data on this issue? Could the Authors show some comparative data on vNOTES versus conventional laparoscopic surgery at their institution? After the operation, prophylactic antibiotics with Cephradine 500 mg every 6 hours for one day and a single dose of Gentamycin 240 mg were given. Could the Authors provide some scientific data on this prophylactic antibiotic regimen?
Author Response
The authors appreciate the reviewer's positive feedback on the manuscript's writing and the relevance of the study's topic. We acknowledge the reviewer's concerns regarding the limitations of a retrospective study and the relatively small sample size, especially in the subgroup analyses. We have addressed these specific points in the revised manuscript and in our responses below.
Specific Comments
1. Length of Stay Discrepancy
Reviewer Comment: The author reported that the patient was typically discharged on post-operative day 1 or day 2. However, the results showed that the mean length of stay was 3.06 days for all patients (2.95 days in the small uterine group and 3.28 days in the large uterine group).
Author Response: We thank the reviewer for pointing out this discrepancy. The statement "The patient was typically discharged on post-operative day 1 or day 2" reflects the standard discharge protocol and the experience of the majority of our patients. In Taiwan, it is a common practice for patients to be admitted one day prior to the operation. Therefore, a discharge two days after the operation would result in an average stay of at least three days. The mean length of stay of 3.06 days was influenced by a small number of patients who had longer hospitalizations, which skewed the average. As stated in the manuscript, we found that a longer length of stay was associated with longer operation times and preoperative anemia. For example, patients with preoperative anemia had an average stay of 3.19 days, with 12 out of 41 patients staying for more than four days.
2. Cervical Amputation
Reviewer Comment: Was cervical amputation always performed in case of a smaller uterus?
Author Response: Cervical amputation was consistently performed in all cases to improve vision and create a larger working space for the surgical procedure. The manuscript states, "We then did cervical amputation and sutured the root with 1-0 Vicryl for better vision and a larger space". This was a standard part of the surgical technique for all patients in this study, regardless of uterine size.
3. Cost-Effectiveness Data
Reviewer Comment: The authors reported that one of the major outcomes of this study was the evaluation of the cost-effectiveness of using an alternative port system. Could the authors provide more data on this issue?
Author Response: We appreciate the request for more specific cost data. Our study focused on the institutional cost-effectiveness and patient affordability of the Lagis LapBase Cap compared to commercial single-port devices. The Lagis LapBase Cap costs approximately 5,500 NTD (about 170 USD) , which is more cost-effective for the institution and more affordable for the patient. In contrast, a typical single-port device costs over 10,000 NTD (310 USD). It is important to note that in Taiwan, the National Health Insurance reimbursement for vNOTES and traditional laparoscopic surgery is the same. The use of this more economical tool may encourage wider adoption of the vNOTES technique, particularly in settings with limited access to commercial single-port devices. While we do not have a detailed breakdown of all costs, the significant difference in the price of the port system itself is a key element of the cost-effectiveness argument presented in our paper.
4. Comparative Data (vNOTES vs. Conventional Laparoscopy)
Reviewer Comment: Could the authors show some comparative data on vNOTES versus conventional laparoscopic surgery at their institution?
Author Response: We thank the reviewer for this suggestion. The primary objective of this retrospective study was to evaluate the outcomes and feasibility of the Lagis LapBase Cap in vNOTES hysterectomy. Since this study was a case series focusing on vNOTES procedures using the Lagis LapBase Cap, we do not have direct comparative data for conventional laparoscopic hysterectomy from our institution. However, our results for surgical time, estimated blood loss, and length of stay are comparable to previous studies from other countries that have compared vNOTES to conventional laparoscopy. For example, a 2015 study by Wang et al. reported that vNOTES was associated with significantly shorter surgical time, less estimated blood loss, and a shorter length of stay compared to traditional laparoscopic hysterectomy.
5. Prophylactic Antibiotic Regimen
Reviewer Comment: After the operation, prophylactic antibiotics with Cephradine 500 mg every 6 hours for one day and a single dose of Gentamycin 240 mg were given. Could the Authors provide some scientific data on this prophylactic antibiotic regimen?
Author Response: We thank the reviewer for this valuable comment. We acknowledge that current guidelines (CDC 2017; ACOG 2018; WHO 2018) recommend a single preoperative dose of cefazolin without postoperative continuation. Our regimen of cefazolin with a single dose of gentamicin, followed by 24 h of cephradine, reflected the institutional protocol during the study period. This approach was based on earlier clinical studies (e.g., Hemsell et al., 1982; Duff et al., 1980s–1990s) when extended prophylaxis was common practice. Later meta-analyses, including the Cochrane review (Smaill et al., 2014), demonstrated no added benefit of prolonged courses. We have clarified this in the revised Methods and added a statement in the Limitations noting the difference between our institutional practice and current guideline-based recommendations
Reviewer 2 Report
Comments and Suggestions for Authors
I have two question: 1 The authors has taken arbitrary decision to use 300 g uterus weight as discriminant value. How they found it? 2 The authors described conversions to laparoscopic hysterectomy. Is it possible to exactly assess time of it in retrospective study based on electronic reports ? The study is long. I think that it would be of value to present some data connected with that time for example learning curve
Author Response
We thank the reviewer for their insightful comments and valuable suggestions regarding our manuscript. We have carefully considered each point and provide the following responses.
1. The discriminant value of 300g for uterine weight
We acknowledge the reviewer's question regarding the choice of 300g as a discriminant value for uterine weight. This value was not chosen arbitrarily, but rather to create two distinct groups for comparative statistical analysis, allowing us to investigate the influence of uterine size on surgical outcomes. In our study, 66 patients had a uterine weight of less than 300g, while 38 patients had a uterine weight of 300g or greater. This grouping was selected to reflect a meaningful clinical distinction and is consistent with similar studies in the literature that have evaluated the impact of uterine size on vNOTES procedures. Our results demonstrated a significant difference in surgical outcomes between these two groups, with the larger uterine group showing longer operation times, more estimated blood loss, and a longer length of stay.
2. Assessment of conversion time and learning curve data
We agree that data on conversion time and the learning curve would be valuable for a comprehensive understanding of the procedure.
-
Conversion Time: As a retrospective study based on electronic medical records, it is challenging to precisely pinpoint the exact time of conversion. We can, however, state that three of the 107 cases required conversion to conventional laparoscopic-assisted vaginal hysterectomy (LAVH) due to factors such as colon adhesion, Cul-De-Sac adhesion, and a narrow vagina. The overall conversion rate was low at 2.8%.
-
Learning Curve: We thank the reviewer for the suggestion to present data on the learning curve. All procedures in this study were performed by a single surgeon, C.S. Hsu, between January 2017 and April 2022. While a formal learning curve analysis was not part of the initial study design, the data collected from this single surgeon's 107 cases inherently captures the evolution of the technique over time. The high procedural success rate of 97.2% and the absence of major complications demonstrate the surgeon's proficiency with the method. We agree that a chronological analysis of the surgical outcomes would be an excellent avenue for future research to formally assess the learning curve